**Data Availability Statement:** All relevant data are within the manuscript.

**Funding:** The author(s) received no specific funding for this work.

# What do Brazilian owners know about canine obesity and what risks does this knowledge generate?

**Mariana Yukari Hayasaki Porsani**[1], **Vinicius Vasques de Oliveira**[1], **Ariane Galdino de Oliveira**[1], **Fabio Alves Teixeira**[1,2], **Vivian Pedrinelli**[1], **Camila Marinelli Martins**[3], **Alexander James German** [4], **Marcio Antonio Brunetto** [1,5]*

1 Veterinary Nutrology Service, Teaching Veterinary Hospital, School of Veterinary Medicine and Animal Science, University of Sao Paulo, Sao Paulo, Brazil, 2 ANCLIVEPA's Veterinary College, Sao Paulo, Sao Paulo, Brazil, 3 Public Health Department, Ponta Grossa State University, Ponta Grossa, Brazil, 4 Institute of Ageing and Chronic Disease, University of Liverpool, Liverpool, United Kingdom, 5 Pet Nutrology Research Center, Nutrition and Production Department, School of Veterinary Medicine and Animal Science, University of Sao Paulo, Pirassununga, Brazil

* mabrunetto@usp.br

## Abstract

Canine obesity is associated with genetic, environmental, and behavioural factors, with the latter including both the behaviour of the dog and the owner. Knowledge about owner perception of canine obesity and its treatment can inform the development of new strategies to help prevent and manage this disease. Therefore, the aim of this study was to evaluate the opinions of dog owners regarding canine obesity and weight management. Dog owners residing in the city of Sao Paulo (Brazil) completed a questionnaire, either at home or in the waiting rooms of 3 veterinary hospitals. Owners determined their dog's body condition score (BCS), and this was compared with BCS determined by a veterinarian. Questionnaire findings from dogs that were in overweight (BCS 6-7/9) or obese (BCS (8-9/9) condition were compared with those in ideal weight (4-5/9) using chi-square tests and odds ratios. A total of 926 dogs were included, of which 480 (52%), 317 (34%) and 129 (14%) were in ideal, overweight and obese condition, respectively. Many owners under-estimated their dog's weight status, with the proportion increasing as the dog's weight status increased (ideal 60/480, 13%; overweight 174/317, 55%; obese 88/129, 68%; $P$<0.001). Although most owners (890/926, 96%) believed that canine obesity could pose health risks, the proportion that disagreed increased as weight status increased (ideal 12/480, 2%; overweight 14/317, 4%; 10/129, 8%; $P$ = 0.006). Finally, although most owners (880/926, 95%) stated that they would let their dog undergo weight management, only a minority (182/926; 20%) believed that a trained professional was needed, and they had various misperceptions including potential cost and what the strategies that would be effective. Based on the findings of this study, it would be advisable for veterinarians to spend time addressing these misperceptions, in the hope of both improving awareness of obesity and the outcomes of weight management.

**Competing interests:** The authors have declared that no competing interests exist.

## Introduction

The pet population is growing year-on-year, with current estimates for the number of dogs in Brazil estimated to be 52.2 million dogs [1]. In Sao Paulo, 50% of households are believed to own at least one dog, and the city's domestic dog population around 2.5 million [2]. The increasing popularity might reflect the strong human-animal bond, with many owners viewing their dogs as members of the family. Dog ownership conveys benefits on owners in terms of physical and mental health [3], with previous studies reporting associations with better arterial pressure control, stress and anxiety reduction, and reduced risk of cardiovascular diseases [4]. However, the close relationship between owner and dog can also have negative consequences such as separation anxiety and obesity [5–7]. Further, an association between treat feeding by owners and overweight condition in their dogs is also reported [5, 8].

A key role of veterinarians is to offer guidance to owners about the most appropriate nutrition, tailored to their dog's needs, including current body condition, concurrent disease, and current medication [9]. Communication between veterinarians and owners must be effective, to ensure that the owner is aware both of the risks of obesity and also the requirements of a weight reduction program including dietary caloric restriction using a therapeutic diet and restricting treats [5, 8, 10, 11].

Owners' perception about canine obesity has already been evaluated in separate studies in Australia [5], the United Kingdom [6], and ten European countries [12]. In Brazil, the profile of owners of dogs in overweight condition has previously been studied at two universities, but body condition score (BCS) was not evaluated [13]. Besides this, there is only one preliminary Brazilian study on the perception of owners about the dog's body condition, but owners were not questioned as for their opinions about obesity [14].

Knowledge of owners' beliefs about pet obesity should help to determine the best way of communicating with owners, it is necessary to identify them on the subject [15]. Therefore, the main aim of the current study was to evaluate the perception of owners from the city of Sao Paulo (Brazil) about both the health risks of canine obesity and also weight loss programs. Furthermore, the study aimed to identify associations between the body condition of dogs and the attitudes of owners towards canine obesity.

## Material and methods

### Ethical considerations

The experimental protocol was conducted according to ethical principles in human and animal experimentation, and was approved both by the Commission on Ethics in the Use of Animals (School of Veterinary Medicine and Animal Science, University of Sao Paulo, protocol number 3443010217) and the Commission of Ethics in Research with Humans (Luiz de Queiroz College of Agriculture, University of Sao Paulo; protocol number 71711317.2.0000.5395).

### Recruitment of dogs and eligibility criteria

Some dogs and owners were recruited as part of a separate study on canine obesity, with owners being approached at their households between November 2017 to November 2018. The remaining dogs and owners were recruited from the waiting rooms of three veterinary hospitals in Sao Paulo: Anclivepa Veterinary Hospital (AVH, public) between August 2018 to November 2018, Oasis Veterinary Hospital (OVH, private) between August 2017 to November 2017, and the Teaching Veterinary Hospital of the School of Veterinary Medicine and Animal

Science of the University of Sao Paulo (TVH-SVMAS) between August 2017 to November 2017.

To be eligible, dogs had to be >8mo age, and could be of either sex, but dogs that could not be handled (e.g. due to aggression), were pregnant, or were in underweight condition (i.e. BCS <4/9) [16] were not eligible. Further, owners had to be ≥18y age, agree to answer all questions, and also to authorise the research team to examine their dog. Finally, people bringing a dog to the hospital on behalf of the actual owner were not eligible.

## Owner questionnaire and BCS assessments

After written consent for the study was obtained, owners were asked to answer a questionnaire (S1 Appendix) regarding their attitudes towards canine obesity and weight loss programs. For this, the veterinarian asked the owners the questions within the questionnaire, which comprised 20 questions in total, 6 of which were in multiple-choice format, whilst the remaining 14 questions required a yes or no answer. If owners had some doubt, the veterinarian (MYHP, FAT, VVO) would clarify the question, taking care not to influence which answer the owner would choose.

Owners were also asked to determine the BCS of their dog. For this, veterinarians first explained to the owners how BCS was classified, and then gave them an illustration of the 9-point scale [16]. The owners then classified their pet's BCS without interference from the veterinarian. Afterwards, one of three trained veterinarians (MYHP, FAT, VVO) then assessed the dog and decided on the BCS separately.

## Statistical analysis

The mean annual caseload seen at each hospital was used to determine the minimum sample size of dogs required from the veterinary hospitals. The Kolmogorov-Smirnov test was first used to check if these data followed a normal distribution, and a sample size calculation was then performed using the simple random method, which assumed a 95% confidence interval, a 5% precision, and an estimated combined prevalence of obesity and overweight of 40.0% in the canine population of Brazil. Based upon this calculation, a minimum of 312, and 233 and 351 dogs were required from AVH, OVH and TVH-SVMAS, respectively.

Data analyses were performed with two computer software packages (JMP version 14.3.0, SAS Institute Inc.; Stats Direct version 3.1.22, Stat Direct Ltd.), and the level of statistical significance was set at $P<0.05$ for two-sided analyses. For the purposes of comparison, dogs were assigned to one of 4 weight categories based upon their BCS: underweight (BCS 1-3/9; n.b. only chosen by some owners but no veterinarians), ideal weight (BCS 4-5/9), overweight (BCS 6-7/9) and obese (BCS 8-9/9). Except where indicated, results are expressed as numbers or proportions with the associated percentage in brackets. Comparisons between the body condition determined by owners and vets were made using the Chi-square test for trend. Either the Cochrane Armitage trend test (where the variable contained only two categories) or the Chi-square test for trend (where the variable contained more than two categories) were used for comparisons between weight category and either owner characteristics, owner attitudes to obesity or owner opinions about weight management. Logistic regression was also used for comparisons amongst the owner attitudes to obesity and weight management and the weight status of their dog. Odds ratios (OR) and 95% confidence intervals (95%-CI) were calculated for dogs in either overweight or obese body condition compared with those in ideal weight (as the reference category). Finally, Chi-squared tests and logistic regression were used to compare differences in owner attitudes to obesity and weight management between those that underestimated the weight status of their dog and those that did not.

## Results

### Final study population

A total of 1070 dogs participated in the study. One hundred and twenty-nine dogs were in underweight condition and were excluded because, with a further 15 dogs excluded because the owners did not complete the full questionnaire. Therefore, data from 926 dogs were included in the final analysis, of which 480 (52%) were in ideal weight, 317 (34%) were in overweight condition and 129 (14%) were classified as obese, according to the veterinarian assessment.

### Owner demographics

The characteristics of the owners participating in the study are shown in the Table 1. In total, 676 (73%) or the owners were female and 250 (27%) were male, with a wide age range and wide range of occupations. Most owners (791, 84%) were educated to high school level or above, and most (789, 85%) had a reported a family income of >\$285.19 USD per month. There were no differences in these variables amongst owners with dogs of weight status ($P \geq 1.000$ for all, Table 2).

### Owner body condition assessments

Although most owners (552/926, 60%) correctly estimated the weight status of their dog, some (52/926, 6%) over-estimated whilst more (322/926, 35%) under-estimated (Fig 1). The proportion of owners that under-estimated their dog's weight status increased as the weight status of the dog increased from ideal (60/480, 13%) to overweight (174/317, 55%), to obese (88/129, 68%; $P < 0.001$). Further, a greater proportion of owners of obese dogs under-estimated the weight status of their dog than did owners of overweight dogs ($P = 0.010$). There were no differences in owner characteristics amongst those that under-estimated their dogs weight status and those that did not ($P > 0.200$ for all, Table 3).

### Owner attitudes to obesity

Owner attitudes about obesity and weight management in dogs are shown in Table 3. Most owners (638/926, 68%) believed that feeding treats could influence weight gain, with no difference in responses when comparisons were made amongst dogs with different weight status ($P = 0.204$). Most owners (870/926, 94%) did not believe that obesity would lead to dogs having more difficulty playing, running, walking or feeling heat, and there were no differences in the responses provided by owners of dogs with different weight status ($P = 0.416$). Most owners (890/926, 96%) believed that obesity could cause risks to a dog's health, but a significant trend existed ($P = 0.006$) whereby more owners disagreed about the health risks as weight status increased from ideal weight (12/480, 2%), to overweight (14,317, 4%), to obese (10/129, 8%). Further logistic regression revealed differences in the attitudes of owners of dogs in obese condition (OR 3.277, CI: 1.838–7.768) but not overweight condition (OR 1.802, CI: 0.822–3.949), compared with owners of dogs in ideal weight as the reference category (Table 3). Further, the odds of an owner disagreeing that obesity could cause risks to a dog's health were greater if they had incorrectly assessed their dog's body condition (Table 4; OR 3.347, 95%-CI 1.706–6.656, $P < 0.001$). However, there were no differences in other attitudes about obesity between owners who did and did not under-estimate their dog's body condition ($P > 0.330$ for all, Table 4).

**Table 1. Relationship between owner characteristics and the weight status of their dog.**

| Characteristic | Total | Ideal weight | Overweight | Obese | P-value |
|---|---|---|---|---|---|
| | **(926)** | **(480)** | **(317)** | **(129)** | |
| **Owner sex** | | | | | |
| Female | 676 (73%) | 339 (71%) | 239 (75%) | 98 (76%) | 0.238 |
| Male | 250 (27%) | 141 (29%) | 78 (25%) | 31 (24%) | |
| **Owner age** | | | | | |
| 18 to 24 years | 85 (9%) | 44 (9%) | 29 (9%) | 12 (9%) | 0.592 |
| 25 to 34 years | 152 (16%) | 84 (18%) | 42 (13%) | 26 (20%) | |
| 35 to 44 years | 185 (20%) | 90 (19%) | 72 (23%) | 23 (18%) | |
| 45 to 59 years | 326 (35%) | 167 (35%) | 111 (35%) | 48 (37%) | |
| 60 to 75 years | 155 (17%) | 80 (17%) | 57 (18%) | 18 (14%) | |
| Older than 75 years | 23 (3%) | 15 (3%) | 6 (2%) | 2 (2%) | |
| **Owner education** | | | | | |
| Did not study | 5 (1%) | 4 (1%) | 1 (<1%) | 0 (0%) | 0.100 |
| From 1$^{st}$ to 4$^{th}$ grade | 63 (7%) | 36 (8%) | 21 (7%) | 6 (5%) | |
| From 5$^{th}$ to 8$^{th}$ grade | 67 (7%) | 37 (8%) | 16 (5%) | 14 (11%) | |
| High school | 320 (35%) | 178 (37%) | 98 (31%) | 44 (34%) | |
| College/University | 390 (42%) | 183 (38%) | 155 (49%) | 52 (40%) | |
| Specialization | 81 (9%) | 42 (9%) | 26 (8%) | 13 (10%) | |
| **Owner income (monthly)** | | | | | |
| No income | 11 (1%) | 5 (1%) | 6 (2%) | 0 (0%) | 0.343 |
| Up to $79.31 | 9 (1%) | 5 (1%) | 4 1%) | 0 (0%) | |
| From $158.63 to $284.94 | 117 (13%) | 66 (14%) | 39 (12%) | 12 (9%) | |
| From $285.19 to $997.80 | 397 (43%) | 202 (42%) | 130 (41%) | 65 (50%) | |
| From $998.0 to 2,428.40 | 236 (25%) | 127 (26%) | 80 (25%) | 29 (22%) | |
| Above $2,428.40 | 156 (17%) | 75 (16%) | 58 (18%) | 23 (18%) | |
| **Job type** | | | | | |
| Government (Public sector) | 60 (6%) | 28 (6%) | 25 (8%) | 7 (5%) | 0.983 |
| Company (private or state) | 219 (24%) | 107 (22%) | 76 (24%) | 36 (28%) | |
| Non-governmental organisation | 14 (2%) | 7 (1%) | 5 (2%) | 2 (2%) | |
| Autonomous | 250 (27%) | 133 (28%) | 84 (27%) | 33 (26%) | |
| Rural property | 5 (1%) | 3 (1%) | 1 (0%) | 1 (1%) | |
| Unemployed | 282 (30%) | 151 (31%) | 94 (30%) | 37 (29%) | |
| Retired | 96 (10%) | 51 (11%) | 32 (10%) | 13 (10%) | |

P-values for sex and job type determined by a 2-by-2 and r-by-c Chi-square tests, respectively; all other P-values determined by the Chi-square test for trend.

## Owner attitudes to weight management

Most owners (880/926, 95%) stated that they would let their dog participate in a weight management plan if necessary, with no difference in attitudes amongst owners who owned dogs of different weight status ($P = 0.762$). Despite this, there was a significant trend for differences in attitudes of owners about whether a trained professional was needed for a weight loss program ($P = 0.038$), with more owners believing that this was not required as weight status increased from ideal weight (374/480, 78%) to overweight (260/317, 82%) to obese (110/129, 85%). However, logistic regression did not reveal differences in the responses of owners when classified according to the weight status of their dog (Table 3). There were also no differences in attitudes to weight management between owners who did or did not under-estimate their dog's body condition ($P > 0.500$ for all, Table 4).

**Table 2. Relationship between owner characteristics and the owner's tendency to under-estimate the weight status of their dog.**

| Characteristic | Owner under-estimated weight status of their dog | | P-value |
|---|---|---|---|
| | **No (664)** | **Yes (262)** | |
| **Owner sex** | | | |
| Female | 477 (70%) | 199 (30%) | 0.204 |
| Male | 187 (75%) | 63 (25%) | |
| **Owner age** | | | |
| 18 to 24 years | 58 (68%) | 27 (32%) | 0.802 |
| 25 to 34 years | 113 (74%) | 39 (26%) | |
| 35 to 44 years | 130 (70%) | 55 (30%) | |
| 45 to 59 years | 236 (72%) | 90 (28%) | |
| 60 to 75 years | 109 (70%) | 46 (30%) | |
| Older than 75 years | 18 (78%) | 5 (22%) | |
| **Owner education** | | | |
| Did not study | 4 (80%) | 1 (20%) | 0.259 |
| From 1st to 4th grade | 48 (71%) | 19 (28%) | |
| From 5th to 8th grade | 241 (75%) | 79 (25%) | |
| High school | 271 (70%) | 119 (30%) | |
| College/University | 57 (70%) | 24 (30%) | |
| Specialization | 48 (71%) | 19 (28%) | |
| **Owner family income monthly** | | | |
| No income | 7 (65%) | 4 (35%) | 0.2972 |
| Up to $79.31 | 8 (89%) | 1 (11%) | |
| From $158.63 to $284.94 | 83 (71%) | 34 (29%) | |
| From $285.19 to $997.80 | 279 (70%) | 118 (30%) | |
| From $998.00 to 2,428.40 | 182 (77%) | 54 (23%) | |
| Above $2,428.40 | 105 (67%) | 51 (33%) | |
| **Job type** | | | |
| Government (Public sector) | 39 (65%) | 21 (35%) | 0.481 |
| Company (private or state) | 152 (69%) | 67 (31%) | |
| Non-governmental organisation | 10 (71%) | 4 (29%) | |
| Autonomous | 191 (76%) | 59 (24%) | |
| Rural property | 3 (60%) | 2 (40%) | |
| Unemployed | 198 (70%) | 84 (30%) | |
| Retired | 71 (74%) | 25 (26%) | |

*P*-values for sex and job type determined by a 2-by-2 and r-by-c Chi-square tests, respectively; all other *P*-values determined by the Chi-square test for trend.

Responses to the question of where owners would seek advice on weight management for their dog are shown in Fig 2. Most indicated that they would approach their veterinary clinic, with no differences between owners of dogs in different weight categories (ideal weight dogs 388/480, 81%; overweight dogs 259/317, 82%; obese dogs 109/129, 85%), whilst only small numbers indicated that they would seek advice elsewhere (including from the internet; dog trainers, pet food industry; family, friends or neighbors; dog breeders; or pharmaceutical companies). However, there were no differences when comparing responses of owners when classified according to the weight status of their dog ($P = 0.815$).

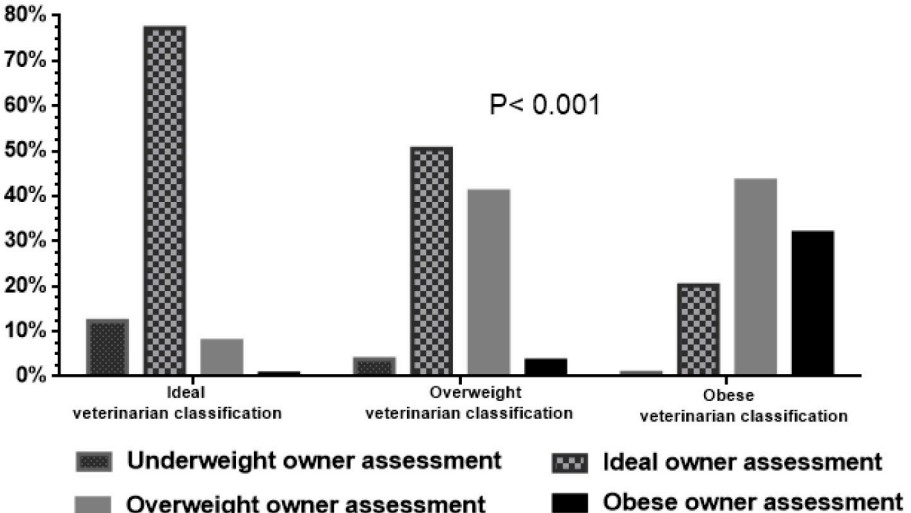

**Fig 1. Comparison of body condition assessments made by 926 dog owners with assessments conducted by veterinarians.** Most owners correctly scored their dog's body condition when the veterinarian judged the dog to be in ideal weight. However, owners commonly under-estimated the condition of dogs judged by veterinarians to be in either overweight or obese body condition (*P*<0.001).

## Owner opinions on success of weight management

Differences were identified amongst owners about which weight loss strategies they thought would be successful in dogs (*P*<0.001). The most popular choices were increasing exercise (768/926, 83%) and using a weight loss diet (661/926, 71%), whilst use of weight loss medications was least popular (76/926, 8%). Compared with use of a weight loss diet as the reference category, more owners believed that exercise would be successful (OR 1.95, 95%-CI 1.56–2.43), whilst fewer owners thought that there would be success using natural products (350/926, OR 0.24, 95%-CI 0.20–0.30), reducing (395/926, OR 0.30, 95%-CI 0.25–0.36) or stopping (306/926, OR 0.20, 95%-CI 0.16–0.24) snacks, using commercial supplements (120/926; OR 0.06; 95%-CI 0.05–0.08), and using drugs (76/926, OR 0.04, 95%-CI 0.03–0.05). Interestingly, more owners thought that decreasing the feeding of treats (395/926, 43%) would be successful than completing stopping treat feeding altogether (306/926, 33%, *P*<0.001). When opinions of owners about the success of different weight loss strategies were stratified according the dog's weight status, no differences were seen (Fig 3, *P*>0.330 for all).

## Owner opinions about hurdles to weight management

Owners expressed a range of opinions as to what difficulties would be faced during a weight loss program, with the cost of treatment (551/926, 60%) and a concern that the dog would be hungry (431/926, 47%) being chosen most frequently. However, there were no differences amongst owners of dogs in different weight categories (Fig 4, *P*>0.145 for all).

## Discussion

The aim of this study was to evaluate owners' perceptions about canine obesity and weight management. In general, most owners believed that obesity could lead to health risks in dogs, although not acknowledging this was more common in owners whose dogs were either overweight or in obese condition. This finding is similar to the findings of previous research

**Table 3. Relationship between the attitudes of 926 dog owners about obesity and the weight status of their dog.**

| Question | Total | Ideal weight | Overweight | Obese | P-value |
|---|---|---|---|---|---|
| | **(926)** | **(480)** | **(317)** | **(129)** | |
| **Do you believe that treats influence weight gain?** | | | | | |
| No | 294 (32%) | 162 (34%) | 96 (30%) | 36 (28%) | 0.354 |
| Yes | 632 (68%) | 318 (66%) | 221 (70%) | 93 (72%) | |
| OR | - - - | - - - | 0.853 | 0.760 | |
| 95%-CI | - - - | - - - | 0.628–1.157 | 0.495–1.667 | |
| **Do you believe that obesity could cause risks to a dog's health?** | | | | | |
| No | 36 (4%) | 12 (2%) | 14 (4%) | 10 (8%) | **0.006** |
| Yes | 890 (96%) | 468 (98%) | 303 96%) | 119 (92%) | |
| OR | - - - | - - - | 1.802 | **3.277** | |
| 95%-CI | - - - | - - - | 0.822–3.949 | **1.383–7.768** | |
| **Do you believe that an obese dog has more difficulty playing, running, walking and feeling more heat than a dog with ideal weight?** | | | | | |
| No | 870 (94%) | 456 (95%) | 292 (92%) | 122 (95%) | 0.416 |
| Yes | 56 (6%) | 24 (5%) | 25 (8%) | 7 (5%) | |
| OR | - - - | - - - | 0.615 | 0.917 | |
| 95%-CI | - - - | - - - | 0.345–1.097 | 0.386–2.179 | |
| **Would you let your dog take part in a weight loss program?** | | | | | |
| No | 46 (5%) | 20 (4%) | 22 (7%) | 4 (3%) | 0.762 |
| Yes | 880 (95%) | 460 (96%) | 295 (93%) | 125 (97%) | |
| OR | - - - | - - - | 1.715 | 0.736 | |
| 95%-CI | - - - | - - - | 0.920–3.198 | 0.247–2.193 | |
| **Do you think a trained professional is needed for a weight loss program?** | | | | | |
| No | 744 (80%) | 374 (78%) | 260 (82%) | 110 (85%) | **0.038** |
| Yes | 182 (20%) | 106 (22%) | 57 (18%) | 19 (15%) | |
| OR | - - - | - - - | 1.293 | 1.641 | |
| 95%-CI | - - - | - - - | 0.903–1.851 | 0.963–2.795 | |

*P*-values determined by Cochran-Armitage test for trend; OR odds ratio determined by logistic regression, comparing dogs in overweight or obese body condition with those in ideal weight (as the reference category) 95%-CI: 95% confidence intervals for OR confidence interval.

[13, 17] and is important because acknowledging this facilitates compliance, which is fundamental to the prevention and treatment of obesity [10].

Owners in the present study overwhelmingly stated that veterinarians were an important source of information about obesity and that involving a trained veterinary professional was important. However, the overall prevalence of overweight and obesity in the current study of 48%, perhaps, indicates that there is disconnect between what the owners say they do and what they actually do. A possible reason for this disconnect is that owners might be unaware that their dog is in an overweight or obese condition. This possibility is supported by our observation that owners of dogs in overweight or obese body condition commonly under-estimated their dog's weight status, with ~41% being discordant with a veterinary assessment of body condition. Similar findings have been seen in previous studies with the proportion of discordance ranging from 44% to 72% [6, 14, 18, 19, 20]. Specifically, Singh et al. (2002) [20] found that 79% of the dogs they studied were overweight, but only 28% of the dogs were considered to be overweight by owners, whilst approximately half the owners of dogs in obese body condition from two Brazilian studies underestimated body condition [13, 18]. There are parallels with parents who tend to under-estimate the weight status of their child [21]. Such owner

**Table 4. Relationship between the attitudes of 926 dog owners about obesity and owner's tendency to under-estimate the weight status of their dog.**

| Question | Owner under-estimated weight status of their dog | | P-value |
|---|---|---|---|
| | **No (664)** | **Yes (262)** | |
| **Do you believe that treats influence weight gain?** | | | |
| No | 212 (32%) | 82 (31%) | 0.853 |
| Yes | 452 (68%) | 180 (69%) | |
| OR | - - - | 0.971 | |
| 95%-CI | - - - | 0.714–1.321 | |
| **Do you believe that obesity could cause risks to a dog's health?** | | | |
| No | 16 (2%) | 20 (8%) | *<0.001* |
| Yes | 648 (98%) | 242 (92%) | |
| OR | - - - | **3.347** | |
| 95%-CI | - - - | **1.706–6.656** | |
| **Do you believe that an obese dog has more difficulty playing, running, walking and feeling more heat than a dog with ideal weight?** | | | |
| No | 37 (6%) | 19 (7%) | 0.334 |
| Yes | 627 (94%) | 243 (93%) | |
| OR | - - - | 0.755 | |
| 95%-CI | - - - | 0.426–1.338 | |
| **Do you think a trained professional is needed for a weight loss program?** | | | |
| No | 536 (81%) | 208 (79%) | 0.646 |
| Yes | 128 (19%) | 54 (21%) | |
| OR | - - - | 0.920 | |
| 95%-CI | - - - | 0.644–1.313 | |
| **Would you let your dog take part in a weight loss program?** | | | |
| No | 31 (5%) | 15 (6%) | 0.505 |
| Yes | 633 (95%) | 247 (94%) | |
| OR | - - - | 1.24 | |
| 95%-CI | - - - | 0.658–2.337 | |

*P*-values determined by Chi-square test; OR odds ratio determined by nominal logistic regression, comparing owners who under-estimated the weight status of their dog with those that did not (as the reference category) 95%-CI: 95% confidence intervals for OR confidence interval.

misperceptions are arguably a challenge for veterinarians when trying to convince owners to start their dog on a weight management program.

Feeding treats and table scraps has been identified as a risk factor for canine obesity in many studies [17, 18, 22]. In the current study, although most owners recognised that treats might be a potential risk for the development of obesity, not acknowledging this risk was associated with a greater odds of their dog being in overweight condition. Arguably, changes in feeding practices, like weighing the daily food amount and controlling treats, are important strategies to prevent the development of excess body weight in dogs [9]; as a result, veterinarians should arguably prioritise the education of owners of overweight dogs about the risks of feeding treats and table scraps.

Successful weight loss in dogs involves the use of therapeutic diets, controlling or reducing intake from of treats, and establishing an exercise routine [23]. In one previous study, owners were asked what they would do to assist their dog in losing weight, with reducing the feeding of treats and changing food type being the main choices [5]. In a second study where the

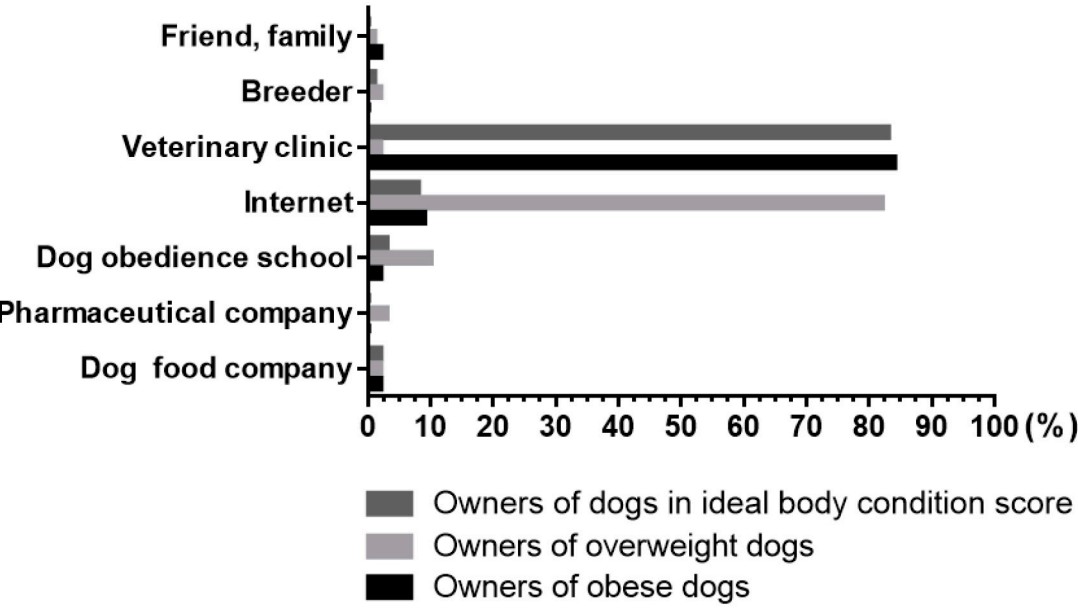

**Fig 2. Opinions of 926 owners about the source of information they would choose for help with weight loss for their dog.**
There was no difference between the responses of owners of dogs in the different weight categories (*P* = 0.815).

owners of overweight and obese dogs were interviewed [13], most indicated that changes in the type and amount of food would be necessary, but only 28% stated that exercise would be important. These findings contrast those of the current study where exercise was chosen about as often weight reduction diet. Such an observation is interesting, because dietary caloric restriction has a far greater effect on weight loss than physical activity [24]. Nonetheless, exercise can help to preserve lean tissue mass [25] and can also contribute to maintenance of body weight and helps prevent obesity [9].

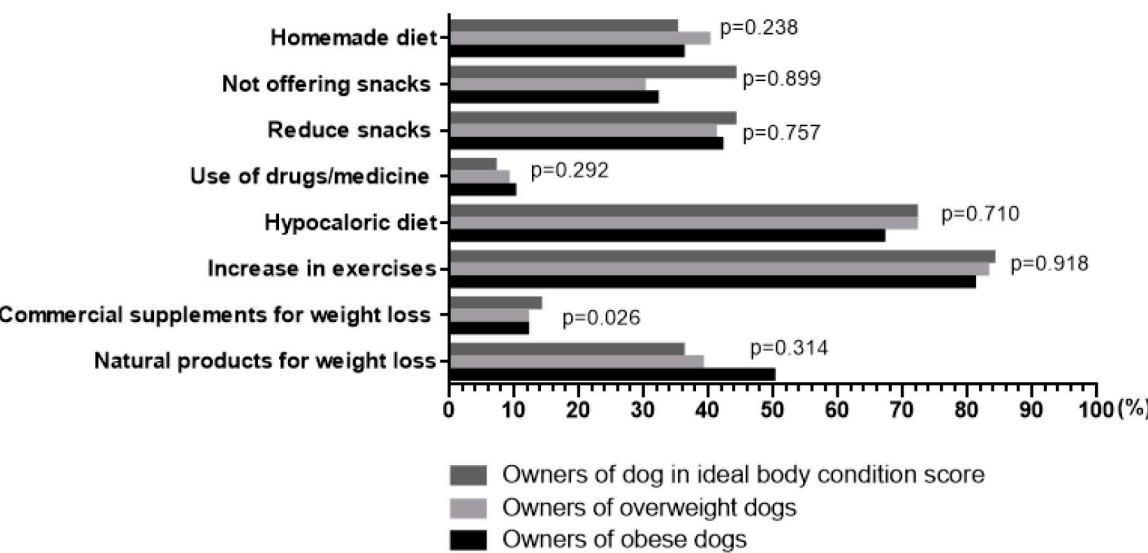

**Fig 3. Opinions of 926 owners about methods that they believed would be successful as a means of inducing weight loss in their dog.**
There were no differences in the responses of owners of dogs in the different weight categories (*P*>0.330 for all).

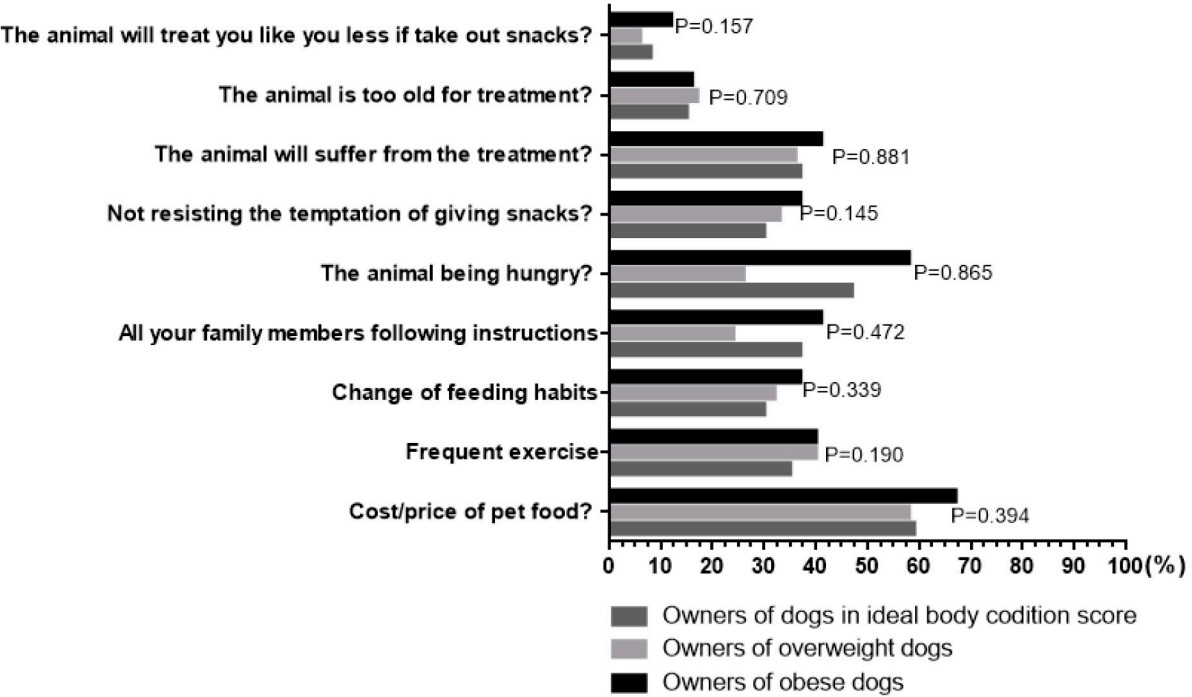

**Fig 4. Opinions of 926 owners about expected difficulties that would be encountered during a weight loss program.** There were no differences amongst the responses of owners of dogs in the different weight categories ($P>0.145$ for all).

In one recent study, the price of food was a key factor for owners of overweight dogs [5, 26], and this was the concern most commonly expressed by owners about weight management. However, such concerns are unfounded in light of the fact that the costs of therapeutic diets during weight loss are cost-neutral on average [27]. Further, the use of a therapeutic weight loss diet is strongly recommended because they improve efficacy of weight management [28], whilst ensuring that essential nutrient intake is maintained despite the caloric restriction [29]. Therefore, once again better education of owners is needed in order that they recognise the importance of using a therapeutic diet for weight management.

Although there is some evidence that compounds such as L-carnitine can affect fat metabolism [30], there is little to no evidence of the effect of dietary supplements on weight loss in dogs [31]. Despite this, most owners expressed the opinion that commercial supplements might be beneficial for weight loss, an opinion more often expressed by the owners of overweight and obese dogs. Once again, this emphasises the need for veterinarians to take time to explain the weight management process to owners, emphasising the key components (such as the use of a therapeutic weight management diet) and also limitations of other options such as supplements.

There are a number of limitations that should be acknowledged, not least the fact that the use of questionnaires may have led to an involuntary bias, because participants altered their answers either to avoid being judged or expressing opinions if they were not certain were correct [9]. Still, it is one of the most common and easy ways to understand owner perception. A further limitation was the fact that overweight and obesity were classified according to BCS, which is not as precise as measurements taken by dual-energy X-ray absorptiometry [28]. However, we used the BCS attributed by the veterinarians as the standard because it was conducted by trained veterinary clinical nutrition team and this method has a good correlation

with more objective alternatives such as dual energy x-ray absorptiometry [32]. Further, we used an illustrative chart of the nine-point BCS system to explain to owners about body conditioning [16]. Although owners believe that having pictures helps them in assigning a score [19], these do not appear to improve the perception of owners regarding the actual weight status of their dog [18–20]. Despite these limitations, the large survey size helped to ensure that the opinions of Brazilian owners regarding canine obesity were properly reflected.

## Conclusions

This is the first Brazilian study regarding the owner´s perception of canine obesity. Many owners under-estimated the weight status of their dog, and this was worst in those with dogs in obese condition. However, most were aware of possible health risks associated with obesity and the need to seek veterinary intervention. Nonetheless, owners had some misperceptions about the weight management process, not least the potential benefit of supplements and concerns over the cost of a therapeutic weight loss diet. Based on these study findings, it is would be advisable for veterinarians to spend time addressing these misperceptions, in the hope of both improving awareness of obesity and the outcomes of weight management.

## Supporting information

**S1 Appendix.**
(DOCX)

## Author Contributions

**Conceptualization:** Mariana Yukari Hayasaki Porsani, Marcio Antonio Brunetto.

**Data curation:** Marcio Antonio Brunetto.

**Formal analysis:** Mariana Yukari Hayasaki Porsani, Ariane Galdino de Oliveira, Fabio Alves Teixeira, Vivian Pedrinelli, Camila Marinelli Martins.

**Investigation:** Mariana Yukari Hayasaki Porsani, Vinicius Vasques de Oliveira, Ariane Galdino de Oliveira, Fabio Alves Teixeira, Camila Marinelli Martins, Marcio Antonio Brunetto.

**Methodology:** Mariana Yukari Hayasaki Porsani, Vinicius Vasques de Oliveira, Ariane Galdino de Oliveira, Fabio Alves Teixeira, Vivian Pedrinelli, Camila Marinelli Martins.

**Project administration:** Marcio Antonio Brunetto.

**Software:** Camila Marinelli Martins.

**Supervision:** Marcio Antonio Brunetto.

**Validation:** Alexander James German.

**Visualization:** Marcio Antonio Brunetto.

**Writing – original draft:** Mariana Yukari Hayasaki Porsani, Vinicius Vasques de Oliveira, Fabio Alves Teixeira.

**Writing – review & editing:** Vivian Pedrinelli, Alexander James German, Marcio Antonio Brunetto.

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
