## [Decision Letter · Decision Letter 0]

13 Aug 2020

PONE-D-20-22044

What do Brazilian owners know about canine obesity and what risks does this knowledge generate?

PLOS ONE

Dear Dr. Brunetto,

Thank you for submitting your manuscript to PLOS ONE. After review by two expert reviewers, they have returned some excellent and highly complimentary comments, but also suggested a few very minor modifications. If you could make these minor modifications and resubmit, the manuscript will be accepted without the need for re-review. Please do not worry about writing a response to reviewers as the comments are so minor. 

Many thanks for submitting your manuscript to PLOS One

It was reviewed by two experts in the field, who returned possibly some of the most complimentary comments I have ever seen on a manuscript review (as an editor and an author)

They have made some very minor comments, mainly grammatical.

If you could make these minor changes and resubmit, your manuscript will be accepted without the need to re-review

As the comments are so minor, please do not worry about a response to reviewers.

I want to commend you on a very nice article, and thank you for submitting it to PLOS One

I wish you all the best with your revisions

Hope you are keeping safe and well in these difficult times

Thanks

Simon

A marked-up copy of your manuscript that highlights changes made to the original version. You should upload this as a separate file labeled 'Revised Manuscript with Track Changes'.An unmarked version of your revised paper without tracked changes. You should upload this as a separate file labeled 'Manuscript'.

We look forward to receiving your revised manuscript.

Kind regards,

Simon Clegg, PhD

Academic Editor

PLOS ONE

2. In your Methods section, please provide additional information about the participant recruitment method and the demographic details of your participants. Please ensure you have provided sufficient details to replicate the analyses such as: a) the recruitment date range (month and year), b) a description of any inclusion/exclusion criteria that were applied to participant recruitment, c) a table of relevant demographic details, d) a statement as to whether your sample can be considered representative of a larger population, e) a description of how participants were recruited, and f) descriptions of where participants were recruited and where the research took place.

3. Please provide additional details regarding participant consent. In the ethics statement in the Methods and online submission information, please ensure that you have specified (1) whether consent was informed and (2) what type you obtained (for instance, written or verbal, and if verbal, how it was documented and witnessed). If the need for consent was waived by the ethics committee, please include this information.

Reviewers' comments:

Reviewer's Responses to Questions

**Comments to the Author**

1. Is the manuscript technically sound, and do the data support the conclusions?

Reviewer #1: Yes

Reviewer #2: Yes

2. Has the statistical analysis been performed appropriately and rigorously? 

Reviewer #1: Yes

Reviewer #2: Yes

3. Have the authors made all data underlying the findings in their manuscript fully available?

Reviewer #1: Yes

Reviewer #2: Yes

4. Is the manuscript presented in an intelligible fashion and written in standard English?

Reviewer #1: Yes

Reviewer #2: Yes

5. Review Comments to the Author

Reviewer #1: Comments on Brazilian dog obesity manuscript

I’ve been reviewing and writing manuscripts for almost 25 years and almost uniquely over that time period this is a manuscript that I find myself struggling to comment on. It is well written, and beyond some small comments listed below, it feels like a published article. I congratulate the authors on a job well done in my opinion. The ability to achieve a critical mass of dogs in this kind of study is extremely difficult at times but achieved here; and the paper reinforces the trends seen in other parts of the world, as the authors note. The fact that the trends seen elsewhere are also exhibited in a different societal region is important knowledge when we think globally about how to challenge the health condition.

Comments.

These are comments rather than changes or suggestions in some ways and the guidance of the editors may be helpful in some of them.

Line 54 – “estimated at…” This is a very precise estimate of dog numbers, now to the individual dog in fact. Yes mathematically true but do we need to either complete the sum for the reader or to these level of accuracy?

Line 136 – why two statistical packages? I am not familiar with these two packages myself – is it a function of their specific statistical tests or merely a difference between authors?

Line 172 – I wondered what the median income in Brazil was to gauge the scale of the $ value here, although there was good information presented on income in the tables further down the manuscript

Line 173, around the tables of demographic information. This is all useful information on the study but I wonder if, given the almost total lack of significance shown, it is just adding volume to the results section here. Could this information be condensed or removed to reduce the length of the section and improve the flow? Should the non-significant p values be indicated as “ns” rather than the specific numerical value.

Reviewer #2: I wish to commend the authors and thank them and the editor for the opportunity to review this paper. It is very rare that I read one as an original submission which is so well written, well analysed and well presented. The tables and figures are excellent and clearly done, the stats look good from my limited knowledge. The information is good and adds to the literature. So overall very well done.

There are a few minor (very minor, sorry) suggestions which are mainly grammatical, but they will only take a few moments and I wouldn’t expect to review this again prior to submission

Once again, thank you, and well done

Line 44- it would (slight reword)

Line 54- you talk about the cities dog numbers- are these domestic dogs as pets which are owned, or strays, and are they all included in one? It seems very precise, so is there a dog owner register? Or would it be easier to say around 2.5 million?

Line 61- Further, an association between treat (reworded and to an)

Line 78- ‘it is necessary to identify their XXXX on the subject’ I think a word is missing where the XXXX is?

Line 114- asked rather than ask

Line 136- not that it makes much difference I doubt, but purely out of interest. Why were two stats packages used? I am guessing it is author preference?

Line 154- compare rather than compared

Quite a number of underweight dogs, and this may be worth a study too as to why they are underweight and if anything can be done to help- just a thought

Line 169- just to clarify- was the owner the person who brought it to the vet? Is it possible that the male and female co-owned a dog? Or the female brought a males dog to the vets etc?

Line 172- is it possible to put some context with the income figure- is that a lot?

Line 303- not sure the word both is needed here

Line 311- is in an overweight or obese condition

Line 329- are important strategies to prevent the development (slight reword)

6. PLOS authors have the option to publish the peer review history of their article (what does this mean?). If published, this will include your full peer review and any attached files.

Reviewer #2: No

---

## [Author Response · Author response to Decision Letter 0]

17 Aug 2020

PONE-D-20-22044

What do Brazilian owners know about canine obesity and what risks does this knowledge generate?

PLOS ONE

Dear Dr. Brunetto,

Thank you for submitting your manuscript to PLOS ONE. After review by two expert reviewers, they have returned some excellent and highly complimentary comments, but also suggested a few very minor modifications. If you could make these minor modifications and resubmit, the manuscript will be accepted without the need for re-review. Please do not worry about writing a response to reviewers as the comments are so minor.

Many thanks for submitting your manuscript to PLOS One

It was reviewed by two experts in the field, who returned possibly some of the most complimentary comments I have ever seen on a manuscript review (as an editor and an author)

They have made some very minor comments, mainly grammatical.

If you could make these minor changes and resubmit, your manuscript will be accepted without the need to re-review

As the comments are so minor, please do not worry about a response to reviewers.

I want to commend you on a very nice article, and thank you for submitting it to PLOS One

I wish you all the best with your revisions

Hope you are keeping safe and well in these difficult times

Thanks

Simon

Dear Editor,

The authors wish to thank you for the attention given to our research. We have considered all the suggestions and comments by the reviewers, which are addressed below. All additional requirements were also met. Thank you once again for considering our work for publishing.

I hope you are also keeping safe and well in these difficult times

Kind regards,

Professor Marcio Antonio Brunetto

Reviewer #1: Comments on Brazilian dog obesity manuscript

I’ve been reviewing and writing manuscripts for almost 25 years and almost uniquely over that time period this is a manuscript that I find myself struggling to comment on. It is well written, and beyond some small comments listed below, it feels like a published article. I congratulate the authors on a job well done in my opinion. The ability to achieve a critical mass of dogs in this kind of study is extremely difficult at times but achieved here; and the paper reinforces the trends seen in other parts of the world, as the authors note. The fact that the trends seen elsewhere are also exhibited in a different societal region is important knowledge when we think globally about how to challenge the health condition.

Comments.

These are comments rather than changes or suggestions in some ways and the guidance of the editors may be helpful in some of them.

Line 54 – “estimated at…” This is a very precise estimate of dog numbers, now to the individual dog in fact. Yes mathematically true but do we need to either complete the sum for the reader or to these level of accuracy?

Author: Thank you, we appreciate your consideration, and based on the review of the other reviewer, we have also improved the sentence.

Line 136 – why two statistical packages? I am not familiar with these two packages myself – is it a function of their specific statistical tests or merely a difference between authors?

Author: Two packages were used by preference and suggestion from the statistical professor who collaborated with us.

Line 172 – I wondered what the median income in Brazil was to gauge the scale of the $ value here, although there was good information presented on income in the tables further down the manuscript

Author: Average per capita household income in Brazil was $ 267.97 in 2019 according to IBGE (Brazilian Institute of Geography and Statistics). Thus, the value found was consistent with the average income of the Brazilian.

Line 173, around the tables of demographic information. This is all useful information on the study but I wonder if, given the almost total lack of significance shown, it is just adding volume to the results section here. Could this information be condensed or removed to reduce the length of the section and improve the flow? Should the non-significant p values be indicated as “ns” rather than the specific numerical value.

Author: As these were not suggestions from the other reviewer (who even made some corrections to these results) we chose to keep the information and p values. As the reviewer also mentions that these are comments rather than changes or suggestions in some ways and the guidance of the editors may be helpful in some of them. We chose to keep it, but we are willing to modify it if you prefer.

Reviewer #2: I wish to commend the authors and thank them and the editor for the opportunity to review this paper. It is very rare that I read one as an original submission which is so well written, well analysed and well presented. The tables and figures are excellent and clearly done, the stats look good from my limited knowledge. The information is good and adds to the literature. So overall very well done.

There are a few minor (very minor, sorry) suggestions which are mainly grammatical, but they will only take a few moments and I wouldn’t expect to review this again prior to submission

Once again, thank you, and well done.

Line 44- it would (slight reword)

Author: We correct, thank you.

Line 54- you talk about the cities dog numbers- are these domestic dogs as pets which are owned, or strays, and are they all included in one? It seems very precise, so is there a dog owner register? Or would it be easier to say around 2.5 million?

Author: Thank you very much, these domestic dogs as pets which are owned. We chose to say "around 2.5 millions".

Line 61- Further, an association between treat (reworded and to an)

Author: We correct, thank you.

Line 78- ‘it is necessary to identify their XXXX on the subject’ I think a word is missing where the XXXX is?

Author: We correct, thank you.

Line 114- asked rather than ask

Author: We correct, thank you.

Line 136- not that it makes much difference I doubt, but purely out of interest. Why were two stats packages used? I am guessing it is author preference?

Author: Two packages were used by preference and suggestion from the statistical professor who collaborated with us.

Line 154- compare rather than compared. Quite a number of underweight dogs, and this may be worth a study too as to why they are underweight and if anything can be done to help- just a thought

Author: We correct, thank you. Thanks for the suggestion, we found it super interesting.

Line 169- just to clarify- was the owner the person who brought it to the vet? Is it possible that the male and female co-owned a dog? Or the female brought a males dog to the vets etc?

Author: Yes, was the owner the person who brought it to the vet. No, only one was considered as the owner of the animals.

Line 172- is it possible to put some context with the income figure- is that a lot?

Author: Author: Average per capita household income in Brazil was $ 267.97 in 2019 according to IBGE (Brazilian Institute of Geography and Statistics). Thus, the value found was consistent with the average income of the Brazilian.

Line 303- not sure the word both is needed here

Author: We correct, thank you.

Line 311- is in an overweight or obese condition

Author: We correct, thank you.

Line 329- are important strategies to prevent the development (slight reword)

Author: We correct, thank you.

---

## [Editor Report · Decision Letter 1]

25 Aug 2020

What do Brazilian owners know about canine obesity and what risks does this knowledge generate?

PONE-D-20-22044R1

Dear Dr. Brunetto,

We’re pleased to inform you that your manuscript has been judged scientifically suitable for publication and will be formally accepted for publication once it meets all outstanding technical requirements.

Kind regards,

Simon Clegg, PhD

Academic Editor

PLOS ONE

Additional Editor Comments:

Many thanks for resubmitting your manuscript to PLOS One

As you have addressed all comments and the manuscript reads well, I have recommended it for publication

You should hear from the Editorial Office shortly

It was a pleasure working with you and I wish you all the best with your future research

Hope you are keeping safe and well in these difficult times

Thanks

Simon

---

## [Editor Report · Acceptance letter]

4 Sep 2020

PONE-D-20-22044R1

What do Brazilian owners know about canine obesity and what risks does this knowledge generate?

Dear Dr. Brunetto:

I'm pleased to inform you that your manuscript has been deemed suitable for publication in PLOS ONE. Congratulations! Your manuscript is now with our production department.

Kind regards,

on behalf of

Dr. Simon Clegg 

Academic Editor

PLOS ONE